# From Vessels to Neurons—The Role of Hypoxia Pathway Proteins in Embryonic Neurogenesis

**DOI:** 10.3390/cells13070621

**Published:** 2024-04-03

**Authors:** Barbara K. Stepien, Ben Wielockx

**Affiliations:** 1Institute of Clinical Chemistry and Laboratory Medicine, Technische Universität Dresden, 01307 Dresden, Germany; 2Experimental Centre, Faculty of Medicine, Technische Universität Dresden, 01307 Dresden, Germany

**Keywords:** embryonic neurogenesis, hypoxia, HIF, vascularization, neocortex, neural progenitor cells, NSC

## Abstract

Embryonic neurogenesis can be defined as a period of prenatal development during which divisions of neural stem and progenitor cells give rise to neurons. In the central nervous system of most mammals, including humans, the majority of neocortical neurogenesis occurs before birth. It is a highly spatiotemporally organized process whose perturbations lead to cortical malformations and dysfunctions underlying neurological and psychiatric pathologies, and in which oxygen availability plays a critical role. In case of deprived oxygen conditions, known as hypoxia, the hypoxia-inducible factor (HIF) signaling pathway is activated, resulting in the selective expression of a group of genes that regulate homeostatic adaptations, including cell differentiation and survival, metabolism and angiogenesis. While a physiological degree of hypoxia is essential for proper brain development, imbalanced oxygen levels can adversely affect this process, as observed in common obstetrical pathologies such as prematurity. This review comprehensively explores and discusses the current body of knowledge regarding the role of hypoxia and the HIF pathway in embryonic neurogenesis of the mammalian cortex. Additionally, it highlights existing gaps in our understanding, presents unanswered questions, and provides avenues for future research.

## 1. Introduction

The neocortex is a folded brain structure that covers most of the human brain surface and plays a major role in our cognitive abilities. With its many gyri and sulci, it constitutes the majority of the cerebral cortex and histologically corresponds to an isocortex with its characteristic six neuronal layers [1,2,3]. It has a recent evolutionary origin, having first appeared in the evolution of mammals [2,4,5,6,7], although homologous structures such as the dorsal cortex and Wulst also exist in reptiles and birds, respectively [8]. In particular, it has undergone a striking degree of radiation since its appearance, resulting in a large diversity of neocortical size and folding complexity between different mammalian species [9,10,11,12,13,14,15]. This suggests an exceptional plasticity of the neocortex, which makes it an important evolutionary novelty that allows for rapid adaptation to versatile functions and diverse ecological niches [4,16].

In humans the neocortex is particularly expanded in comparison to other mammals, including closely-related apes [12,17], and as such has long been an object of particular focus for neurobiologists. Functionally, it has been credited with facilitating much of the human cognitive achievements in both intellectual and social domains, including learning, memory, speech and emotional regulation [18,19,20]. Therefore, it is not surprising that any perturbations in its development, which occurs primarily pre- and perinatally [12,21], have profound clinical consequences [22,23,24,25,26]. Particularly, common gestational and obstetric complications such as intrauterine growth restriction or prematurity often result in inappropriate tissue oxygenation during critical developmental phases [27,28,29]. Such changes in oxygen pressure, both during neurogenesis and during subsequent events, which include neuron migration, synaptogenesis or myelination may cause long-lasting neurodevelopmental and neuropsychiatric repercussions [23,27,28]. A prerequisite for addressing pathologies related to these processes in a clinical setting is understanding the physiological role of oxygen pressure and its downstream effects, typically mediated by the HIF signaling pathway, on various cell types in the developing neocortex. In this review, we present the current state of knowledge on the role of oxygenation and HIF signaling in embryonic and perinatal cortical development.

## 2. Neurogenesis in the Prenatal Neocortex

At the histological level, the mature neocortex is a highly ordered structure in which six distinct neuronal layers can be distinguished [1,2,3]. It arises from the dorsal telencephalon during embryonic development [30]. First, a single cell layer neuroepithelium composed of symmetrically dividing neural stem cells (NSCs) folds up and closes to form a neural tube (NT) [31,32] (Figure 1A,B). This morphogenetic process leads to a change in tissue arrangement so that the apical surface of the neuroepithelium lines a fluid-filled ventricle inside of the NT and its basal side faces the outside. This structure will give rise to the entire central nervous system (CNS) in the form of the brain and spinal cord. Embryonic NSCs produce almost all CNS neurons and glial cells, with an exception of microglia [33,34,35,36]. At the onset of neurogenesis proliferating neuroepithelial cells (NECs) transform into radial glia (RG) and begin to divide asymmetrically in order to both self-renew and give rise to more differentiated neural progenitor cells (NPCs) and eventually neurons [33,36,37,38,39,40] (Figure 1C).

In the embryonic neocortex NPCs reside and divide in two adjacent germinal zones. The primary zone, called the ventricular zone (VZ) due to its direct contact with the cerebrospinal fluid (CSF)-filled ventricles on the apical side, contains apical radial glia (aRG). Characteristic features of aRG include (1) cell morphology with an apical and a basal process providing direct contact to both ventricular and pial surfaces, (2) intrakinetic nuclear migration, during which the nucleus-containing soma of the RG cell migrates between the apical and basal most borders of the VZ according to its cell cycle with mitosis occurring at the ventricular surface, and (3) high expression of marker genes such as nuclear Pax6 (Paired box 6) and Sox2 (sex determining region Y-box 2) [36,37,38,39,41,42]. The asymmetric divisions of these cells typically give rise to more differentiated progenitor types, which delaminate from the apical surface, migrate basally and settle within the second germinal zone, referred to as subventricular zone (SVZ) [37,42,43,44,45,46,47,48,49,50,51]. These progenitors, collectively described as basal progenitors (BPs), can be subdivided into various classes depending on their cell morphology, marker expression and proliferative potential. The two most prominent types are basal intermediate progenitors (bIPs) and basal radial glia (bRG). bIPs have no direct contact with either ventricular or pial surface and show a limited proliferative potential, typically dividing only once to produce two neurons [37,38,42,52]. They are characterized by the expression of the nuclear marker Tbr2/Eomes (T-box brain protein 2/Eomesodermin) [41]. In contrast, bRG retain the basal process, show a marker gene expression more similar to the parental aRG, and are capable of self-renewal through asymmetric cell divisions [45,46,47,48,49,53]. Neurons born from NPC divisions undergo radial migration toward the pial surface along the radial fibers of the aRG cells [38,54,55]. The formation of the six-layer cortex depends on a precise timing of birth and an inside-out migration pattern of newly-born neurons, in which later-born neurons migrate past earlier-born neurons to settle in the upper layers of the forming cortical plate (CP) [56,57,58,59,60,61,62] (Figure 1D). This process is orchestrated by an early born transient population of Cajal-Retzius neurons residing in the basal-most marginal zone (MZ). They produce and secrete reelin, whose gradient acts as a guidance cue for neuronal migration [23,63]. This neuronal population, along with another transient population of early-born subplate (SP) neurons at the apical side of the CP, undergoes apoptosis at the end of development [23,63,64].

While small mammals, represented by the majority of rodents including the laboratory model species mouse and rat, have relatively simple and smooth (lissencephalic) cortices, most other mammals have larger cortices with variable degrees of folding (gyrencephaly) [12,13,17]. This feature is thought to arise from the evolutionary expansion of the SVZ in large-brained species and the abundance of more proliferative NPCs in this zone, particularly the bRG [16]. As an example, humans have a complex SVZ that can be subdivided into an inner and outer part (iSVZ and oSVZ respectively) [65]. It contains a larger proportion of self-renewing progenitors such as bRG than the narrower mouse SVZ, in which the neurogenic bIPs are the dominant BP population [37,42,50]. The length of the neurogenic period, during which cortical pyramidal neurons are produced, varies in different mammalian species and is directly correlated to the size of the adult neocortex [12,66,67]. Depending on the species, neurogenesis is either followed by, as in the house mouse, or may partially overlap, as in humans, with gliogenesis, which provides cortical astrocytes and oligodendrocytes [68,69,70,71,72,73,74,75,76,77]. It is also noteworthy, that the local germinal zones are primarily responsible for the generation of glutamatergic excitatory neurons, while the majority of inhibitory interneurons are generated in the ventral regions of the developing brain, mainly in the lateral and medial ganglionic eminences (LGE and MGE) [2,78,79,80]. These neurons enter the neocortex via tangential migration [2,78,79,80]. The embryonic NPCs also contain a subpopulation of dividing cells that persists beyond the perinatal period and give rise to the adult progenitor cells of the subependymal SVZ [81,82] (Figure 1E).

Any defect in the establishment of the stereotypical cortex microarchitecture has profound functional consequences and can lead to clinical symptoms, such as those present in intellectual disability or seizure disorders [24,83,84]. These perturbations may arise from intrinsic developmental impairments such as mutations in neurogenesis-related genes [24,83,85,86] but more commonly stem from the external environment, particularly nutrient deprivation or infections [87,88,89,90,91]. An efficient delivery of nutrients, including the metabolically necessary oxygen and glucose, requires a tightly spatiotemporally-regulated ingrowth of blood vessels. The pattern of neocortical vascularization and its interplay with embryonic neurogenesis as well as the role of oxygen-sensing HIF signaling pathway, will be discussed in the next chapters.

## 3. Physiological Hypoxia in the Developing Brain

While the atmospheric oxygen concentration, often referred to as normoxia, fluctuates around 21%, corresponding to an oxygen partial pressure (PO_2_) of approximately 21 kPa (or 159 mmHg), virtually all tissues in the human body experience oxygen levels significantly lower than this. Brain tissue oxygenation under homeostatic conditions reaches around 4.67–5.33 kPa (35–40 mmHg), which is almost three times lower than the oxygen tension in arterial blood [92]. In the adult rat brain, O_2_ pressure in the cortex varies from 2.53 to 5.33 kPa (19–40 mmHg) in cortical grey matter and from 0.8 to 2.13 kPa (6–16 mmHg) in white matter, similar to that measured in other mammalian species [92,93]. It can reach even lower levels in deeper brain regions e.g., 0.55–1.07 kPa (11–16 mmHg) in the midbrain. The highest oxygen levels in the adult brain have been measured in the ventricles and the hippocampal dentate gyrus (DG) [94]. Even the pial surface experiences only around 8% O_2_ level. Under systemically induced hypoxia, e.g., at high altitudes, or under pathological conditions, oxygen pressure in the brain can be further lowered despite physiological compensation in breath and heart rate, enhanced erythropoiesis and cerebral blood flow [94,95].

Importantly, most of the early embryonic development also occurs under relatively hypoxic conditions, including in the forming CNS [96,97,98]. The degree of physiological hypoxia varies between different brain regions and at different developmental stages or even in subregional zones within embryonic brain structures [99,100]. For example, in the developing midbrain the oxygen tension in the dopaminergic neurogenic niche restricted to the VZ area around the aqueduct is below 1.1% [99], whereas in other fetal brain regions, oxygen levels are typically in the range of 2–5% O_2_ [101]. Embryonic and adult brain germinal zones are thought to be physiologically hypoxic with estimated 18–24 mmHg oxygen pressure (2.5–3% O_2_) in the fetal SVZ and 8–48 mmHg (1–6% O_2_) in the adult SVZ respectively [100,102].

Deviations from physiological hypoxia in either direction are associated with abnormal development [96]. Pathological hypoxia can be defined by oxygenation levels below what is physiological for a given tissue and developmental stage. The exact ranges for physiological and pathological hypoxia vary by developmental stage, cell type and tissue architecture in relation to local blood vessel density. In addition, increased O_2_ levels beyond the physiological set point may also have adverse effects. Rat embryos cultured in high oxygen (45%) showed reduced amounts of apoptosis accompanied by morphogenetic defects in otic vesicle invagination, NT closure, somite formation and embryo turning [96]. Reducing the oxygen below physiological levels had a less dramatic effect, causing mostly growth retardation and increased apoptosis. During embryonic and fetal development, non-physiological hypoxia can result from placental defects, obstruction of blood flow within the fetoplacental unit, cardiovascular malformations of the embryo, and a variety of other causes [27,103]. Preterm birth, a common obstetric complication, can cause both pathological hypoxia of the newborn due to lung underdevelopment, and hyperoxia [27,28,29]. The latter is caused by higher than physiological levels of oxygenation in a tissue and is typically achieved by life-saving oxygen treatments [28,29]. Overall while O_2_ is a vital metabolic growth factor, increasing or limiting its levels has effects on development beyond energy production. Understanding the role of physiological hypoxia at different stages of development is critical for designing strategies to mitigate the negative consequences of non-physiological changes in oxygen pressure.

## 4. Oxygen-Sensing and the HIF Signaling Pathway

Given the critical role of oxygen in cellular metabolism, it is not surprising that organisms have evolved robust oxygen sensing mechanisms that allow them to adapt to changes in oxygen concentration at both systemic and local tissue levels. In vertebrates, cellular oxygen sensing is mediated primarily by the HIF signaling pathway [104,105,106,107,108]. The molecular components of this pathway are expressed in virtually all cells of the body and are remarkably conserved in evolution [107]. The key players encompass the heterodimeric hypoxia-inducible factors (HIFs), which are transcription factors belonging to basic helix-loop-helix PAS (Per/Arnt/Sim) protein family [107,109,110] (Figure 2A). These heterodimers consist of a constitutively expressed nuclear subunit HIF-1β also known as ARNT (aryl hydrocarbon receptor nuclear translocator) and either the HIF-1α or HIF-2α subunit [104,105,107]. Both HIF-1α and HIF-2α proteins share a similar domain structure but differ in their expression pattern [107,111,112,113,114]. They consist of a N-terminal bHLH (basic helix-loop-helix) domain responsible for DNA binding followed by two PAS dimerization regions, an ODD (oxygen-dependent degradation) domain and C-terminal transactivation domain (TAD) [107,111,112,115]. HIF-1α is ubiquitously expressed, whereas HIF-2α is more restricted [113,116,117,118,119,120]. Moreover, despite their similar structure, they are not functionally equivalent. Expression of HIF-2α from a HIF-1α locus leads to embryonic and extraembryonic tissue patterning defects, defective hematopoietic stem cell differentiation, and promotes overproliferation at the expense of differentiation due to increased expression of pluripotency genes such as Oct-4 (Octamer-binding transcription factor 4) [121].

HIF-1/2α subunits are continuously translated from mRNA and in the presence of sufficient oxygen pressure undergo immediate degradation in the cytoplasm [122,123]. This degradation is facilitated by a group of enzymes in the 2-oxoglutarate-dependent oxygenase superfamily known as the prolyl 4-hydroxylases (PHD), which catalyze the hydroxylation of the prolyl side chain of HIF-1/2α proteins [107,124,125]. Three members of this group (PHD1, PHD2 and PHD3) have been described in vertebrates [105,107,125]. PHD2 is the only one of the three that is required for survival as its deletion causes embryonic lethality between E12.5–E14.5 [105,126]. The activity of the PHD enzymes requires molecular oxygen as well as several cofactors, namely 2-oxoglutarate, ascorbate and Fe^2+^ ions [107,127]. Proline hydroxylation of HIF-1/2α targets them for ubiquitination by the Von Hippel-Lindau tumor suppressor (VHL) E3 ubiquitin ligase protein resulting in their proteasomal degradation [128,129]. HIF-1α can additionally be regulated by another enzyme, asparaginyl hydroxylase HIF1AN (FIH-1), which hydroxylates an asparagine residue in its C-terminal transactivation domain (TAD) resulting in decreased transcriptional activity [130,131,132,133]. Upon reduction of tissue oxygen levels PHD enzymes gradually lose their activity and non-hydroxylated HIF-1/2α proteins translocate to the nucleus where they dimerize with HIF-1β subunits to form functional transcription factors [104,105,106]. They bind to conserved DNA sequence motifs described as hypoxia response elements (*HRE*s) in promoter regions of a great number of target genes, involved in many different processes including glycolysis, iron metabolism, erythropoiesis and angiogenesis [105,134,135,136]. Canonical targets include angiogenesis-promoting *Vegf* (vascular endothelial growth factor), erythropoiesis stimulating *Epo* (erythropoietin) and *Pgk1*, a glycolysis enzyme (phosphoglycerate kinase 1) [105,123,137]. Consistent with their functional non-equivalence, HIF-1α and HIF-2α-containing complexes regulate overlapping but not identical sets of targets [114,138]. The regulation of HIF complex assembly is depicted schematically in Figure 2B.

In addition to HIF-1/2α, vertebrate genomes encode for the third member of the HIF transcription factor family, HIF-3α [111,112] (Figure 2A). The HIF-3α protein levels can also be regulated by hypoxia and the PHD/VHL-mediated degradation [111,112]. Prenatally, it is expressed during late embryogenesis in mice, predominantly in the lung and heart, but also in the developing brain. HIF-3α differs from HIF-1/2α in its protein structure in that it lacks the C-terminal TAD. Instead, the gene encodes a leucine zipper (LZIP) domain in its C-terminal portion [111,112]. However, HIF-3α can be expressed as multiple isoforms with different domain inclusions, corresponding to different expression patterns and functions [112]. It is widely regarded as a dominant negative regulator of HIF-1/2α-driven transcription at least partly due to the ability of some of its isoforms to act as competitive inhibitors. Selected forms of HIF-3α are able to bind DNA and dimerize with either HIF-1β or other α subunits to form complexes that lack transcriptional activity. Interestingly, HIF-3α gene is also a target of HIF-1/2α-mediated transcriptional activation thus providing a negative feedback loop. In addition to its inhibitory function, HIF-3α can also induce transcription of a number of genes distinct from other HIF-α proteins by an incompletely understood mechanism [111,112].

The precise pattern of activation and downstream effects of the HIF pathway depends on the cellular and tissue context. In both adult and prenatal settings, certain tissues and areas experience physiological hypoxia resulting in constitutive HIF pathway activation [101,123]. Adult mice reared in normoxia (21% atmospheric O_2_) produce detectable HIF-1α protein levels in the brain, kidney, liver, skeletal muscle and heart but not in the lung [123]. HIF pathway activity is also required for proper development as global constitutive knockout of HIF-1α in mice causes growth retardation by E8 and embryonic lethality by E10.5 [139,140,141]. These phenotypes are primarily caused by defective angiogenesis and cardiovascular malformations, leading to widespread hypoxia and cell death. HIF-1α KO embryos also display a failure of neural-fold closure accompanied by decreased embryonic PGK expression, which is particularly pronounced at the neural fold margins [140]. Perhaps not surprisingly, these phenotypes largely overlap with the effects of embryonic hyperoxia highlighting the critical role of oxygen-driven regulation of the HIF pathway for developmental processes [96]. This points to a specific role of HIF signaling in CNS development that is at least partially independent of its role in vascularization.

## 5. Cortical Angiogenesis and Its Interaction with NPCs in the Embryonic Neocortex

Considering that the oxygen delivery to embryonic organs depends on the development of the nascent vasculature, the pattern of local tissue hypoxia and HIF pathway activation changes according to the spatial and temporal progression of angiogenesis. In the early stages of development, the cerebral cortex is avascular and progenitor cells divide under physiological hypoxia [96,98,142]. The dorsal telencephalon of a mouse is initially completely free of blood vessels (Figure 1B). Pial vessels, representing the venous side of the circulatory system, are the first to arrive and cover the basal surface of the future neocortex by E9 in mouse [143] or six weeks of gestation in humans [144]. Initially, these vessels form a perineural vascular plexus (PNVP) [145]. This phase of surface vascularization is followed by internal vascularization when the PNVP generates sprouts, which penetrate the pia, invade the embryonic murine cortex and grow radially in the direction of the ventricular surface [145,146] (Figure 1C). Later the radial vessels also produce horizontal branches [145,147]. Meanwhile, periventricular vessels, which form a second plexus near to the apical (ventricular) side, are initially present only in the ventral telencephalon from where they gradually sprout into the dorsal telencephalon between E10-11 [143]. Around E12.5 a honeycomb-like vascular plexus (periventricular vascular plexus; PVP) forms near the ventricular side in the SVZ, just above the VZ [145,146,148,149], which also connects both periventricular and pial vessels [148] (Figure 1D). Gradually, the pial and periventricular plexi form connections that are thought to approximately reflect the arterial (periventricular)-to-venous (pial) flow. The lower region of the SVZ contains a particularly extensive plexus, while most of the VZ except for its uppermost zone remains avascular. This avascular region decreases in size during development and eventually disappears postnatally, which correlates with the decrease in VZ thickness [148]. At E14.5, a secondary internal plexus forms just under the CP and both plexi connect via penetrating branches [146]. The forming CP contains primarily vertical vessels with SP forming a boundary to the honeycomb plexus of the VZ/SVZ/IZ. This ventral-to-dorsal gradient of angiogenesis is similar along the entire rostrocaudal span of the forebrain [143]. This pattern is also remarkably conserved in evolution with species such as the softshell turtle, mouse and ferret exhibiting similar vascular architecture [148]. In the perinatal period, starting around E18.5 a more homogeneous vascularization pattern forms with dense parenchymal capillaries [146].

The gradient of angiogenesis in the dorsal telencephalon precedes that of neurogenesis [60,143,149,150,151]. Angiogenic sprouting into the cortical parenchyma relies predominantly on VEGF signaling associated with local hypoxia [139,142]. *Vegf* is a target of HIF, which serves as a guidance molecule for endothelial cell (EC) migration and proliferation [152]. Both VEGF and HIF-1α protein expression is high in the early VZ and decreases during development [148]. Exposure of pregnant mouse dams to systemic hypoxia by placing them in a 10% O_2_ atmosphere increased both HIF-1α expression in the embryonic VZ as well as vascular branching, while NPC-specific HIF-1α deletion severely impaired angiogenesis [148]. Interestingly, PVP-specific angiogenesis was also shown to be driven, at least in part, by a cell-autonomous function of homeobox transcription factors expressed by invading ECs [143]. Notably, the same transcription factors are also expressed by the surrounding NPCs: the ventral ECs express the ventral progenitor markers Nkx2.1 and Dlx1/2, while dorsal periventricular ECs express the dorsal telencephalon aRG marker Pax6. None of these transcription factors are produced by the pial vessels and their deletion causes regional vascularization defects by affecting the expression of BDNF (brain-derived neurotrophic factor), VEGF-A and its receptors (Kdr/Flk1 and Flt1) in ECs. Surprisingly, this mechanism appears to be independent of the classical hypoxia-sensing HIF pathway [143]. In the same time, perturbation of neuronal layering e.g., in the reeler mouse has little effect on the vascularization pattern [146]. In contrast, early postnatal vascularization in the mouse forebrain requires non-EC autonomous VEGF and HIF-1/2α signaling [153]. Deletion of PHD2 specifically in postmitotic neurons during the perinatal period increased both HIF-1/2α protein levels and vessel density, branching and EC proliferation, while neuronal HIF-1/2α KO had an opposite effect. Manipulation of HIF-1/2α levels in neurons affected not only cell-autonomous VEGF expression but also its production by astrocytes by a yet unknown mechanism. Moreover, VEGF is not the only HIF-dependent angiogenic factor. Another HIF target, adrenomedullin (Adm), additively increased VEGF-A-induced endothelial sprouting in a MAP2K1/1, Src and PI3K signaling-dependent manner. Exposure to systemic hypoxia (8% O_2_ atmosphere) increased both *Adm* and *Vegfa* transcription in the adult brain.

Studies of temporal changes in the developing vessel architecture of the murine neocortex revealed specific spatial relationships with resident progenitor cells, their progeny as well as cells migrating from other brain regions [146,148,149]. The spatiotemporal pattern of angiogenesis in the developing neocortex establishes hypoxic and perivascular zones that serve as niches for NPC subtypes [146,148,149]. Notably, aRG cells occupy the most hypoxic region of the VZ and their somata are located away from blood vessels [142,148,154]. Increasing vessel density in this region by genetic manipulation, for instance by heterozygous KO of Flt1 (Fms related receptor tyrosine kinase 1), a VEGF receptor, or by systemic hypoxia reduces the number of aRG. At the same time, the number of BPs increases, likely due to the progenitor division mode being pushed towards differentiation [148]. Conversely, homozygous HIF-1α deletion in NPCs resulted in profoundly impaired neocortical angiogenesis. Nevertheless, the number of mitotic BPs was increased at the expense of aRG cells suggesting that HIF-1α is required to prevent premature neurogenesis independently of its effect on vessel ingrowth.

The correlation between the progression of neurogenesis and the progressive vascularization of the developing neocortex has prompted studies on the potential role of direct cell-cell interactions between the vasculature and neuronal progenitors. In-growing vertical blood vessels in the CP follow the direction of RG fibers and ECs are in direct physical contact with these fibers through gaps in pericyte coverage [147]. These interactions affect vessel stabilization after mid-neurogenesis by inhibiting Wnt pathway activity in ECs [147]. Direct contacts between NPCs and ECs were also observed in the VZ. The Flk1^+^ tip cells of sprouting new vessels contact mitotic aRG cells via long filopodia and the frequency of these contacts changes as the neurogenesis progresses both in the dorsal and ventral telencephalon [148,155]. These interactions regulate the proliferation/differentiation balance of the apical but not basal progenitors in both mouse and human and aRG marker expression is increased by these contacts [148,155]. A high density of EC filopodia extends the mitotic phase of progenitors and thus the total cell cycle length, resulting in more differentiative divisions that produce BPs and neurons [155]. In turn, the growth of filopodia is induced by VEGF-A secreted by mitotic apical NSCs. An endothelial deletion of the VEGF receptor and negative angiogenesis regulator KDR/Flk1 (kinase insert domain receptor) enhances angiogenesis and reduces HIF-1α protein expression as well as aRG maintenance while increasing differentiation into BPs [148]. Interestingly, mouse ventral but not dorsal telencephalic apical progenitors associate their basal process with periventricular vessels [156]. aRG cells in both regions initially extend their basal processes to the pial surface, but ventral progenitors progressively lose these contacts between E11.5 and E16.5. Instead, they establish interactions with periventricular vessels mediated by laminin-containing basement membrane. This contact is maintained throughout the cell cycle, including mitosis. Its loss by ITGβ1 deletion in RG cells led to fewer mitotic divisions, resulting in defects in cortical interneuron production and functional consequences in circuit formation. PV^+^ (parvalbumin) and SST^+^ (somatostatin) interneurons were underproduced which led to diminished synaptic inhibition in the neocortex [156]. Interestingly, human but not mouse aRG cells in dorsal telencephalon also lose their pial contact in later developmental stages and their basal processes were shown to frequently contact blood vessels [157]. Whether these contacts mediate similar effects on aRG cell division to those in ventral telencephalon remains unclear. Given the greater tissue thickness of mouse ventral telencephalon in comparison to dorsal telencephalon during late neurogenesis, as well as the expansion of dorsal telencephalon thickness in human evolution, the transfer of basal process contact from pia to blood vessels may reflect an adaptation to a general limit on the process length. Another possibility is that this morphological difference is linked with the ability of human dorsal progenitors to generate inhibitory interneurons, a function normally restricted to ventral progenitors in rodents [158].

The ingrowth of blood vessels into the developing cortex relieves hypoxia and coincides with NSC differentiation, the appearance of Tbr2^+^ BPs, and the generation of neurons in both mice and ferrets [142]. During cortical neurogenesis mitotic intermediate progenitor cells (IPCs) in the SVZ and IZ, particularly Tbr2^+^ BPs, were shown to preferentially reside in the vicinity of blood vessels and divide adjacent to vascular branch points in areas covered by pericytes [146,148,149]. This pericyte contact was associated with suppression of neuronal differentiation [148]. Some VZ Tbr2^+^ cells in the VZ, representing delaminating BPs, also had direct contact with endothelial tip cells and Tbr2^+^ cell density between E12-14 increased in a lateral-to-medial gradient correlating with the gradient of angiogenesis [149]. Manipulation of NPC identity by overexpression of Tbr2 to induce BP formation reduced vascular branching, while overexpression of Notch intracellular domain (NICD) to promote aRG maintenance increased it. Because overexpression of these proteins regulated *HRE*-mediated transcription in opposite ways it is likely that they directly or indirectly modulate HIF-1α activity [148]. These studies highlight the crucial role of the activation of hypoxia-mediated HIF pathway in coordinating vessel in-growth and progenitor self-renewal/differentiation balance. This interdependence between hypoxia, HIF pathway activation and angiogenesis-mediated hypoxia relief also makes it challenging to experimentally disentangle and pin-point cause-and-effect relationships between these factors.

Gliogenic progenitors are also affected by local vascular architecture. Oligodendrocyte progenitor cells (OPCs), similar to neurogenic BPs, reside in the proximity of honeycomb vessels in upper IZ and SP albeit in lower vessel density regions than BPs [148]. They also prefer vessel branching points but are adjacent to ECs and avoid direct contact with pericytes. OPCs cultured on ECs in vitro displayed a less differentiated mRNA expression pattern with lower levels of mature markers Mbp and S100β and higher level of PDGFRα [148]. Progenitor cells are not the only cell types affected by cortical vascularization. Neurites of newly born migrating neurons also preferentially align with blood vessels [146], which have been shown to guide the migratory pattern of ventrally born interneurons destined for the neocortex [159,160,161,162]. Collectively, changes to cortical angiogenesis patterns during neurogenesis influence neuron and glia production as well as neuronal survival eventually resulting in altered cortical thickness [148,155,163]. The effects of vasculature are not limited to direct cell-cell contacts or oxygenation, as they also encompass the delivery of other signaling molecules and nutrients. As an example, low glucose, which in vivo results from low vessel density, decreases proliferation and increases differentiation of fetal-derived (E15.5) mouse NSCs in vitro [164].

## 6. The Role of Hypoxia and HIF Pathway in Cultured NPCs

In addition to the direct effects of cell-to-cell contact, cortical vascularization indirectly affects resident progenitors and their progeny by providing diffusible components to the stem cell niche. One of the most important indirect effects of increased vascular density is an elevation of local tissue oxygenation and the relief of physiological hypoxia. These affect the cells residing in the niche primarily through the modulation of the HIF signaling pathway, as described in detail below.

The influence of oxygen directly as a metabolic substrate or via its effect on HIF pathway activation on the survival, proliferation and differentiation of NPCs has been extensively studied using in vitro cultures. These have the advantage in terms of being able to tightly control the O_2_ concentration, however, suffer from the lack of native tissue context. Multiple observations show that hypoxic conditions in a physiological range for a developing fetal brain and HIF-1/2α stabilization promote stemness and suppress precocious differentiation of NSCs and NPCs in culture [102,137,165]. This seems to be an overarching principle for many other stem cell types, including human ESCs [166,167], which maintain higher levels of pluripotency under 3–5% compared to 21% O_2_ [168,169]. However, low oxygen levels (2% O_2_) have also been shown to alter germ layer specification of hESCs by promoting a more neural fate in a HIF-1α and -1β dependent manner [170]. Spontaneously differentiating ESCs in normoxic culture with no detectable HIF-1α protein maintained higher expression of pluripotency markers and sustained low Sox1 levels [171]. As these ESCs aggregates expanded, they spontaneously generated hypoxic zones leading to HIF-1α stabilization and neural lineage commitment. These changes were dependent on HIF-1α, as its knock-down led to reduced expression of neuroectodermal and neural genes, such as Sox1, Nestin or Pax6 while increasing pluripotency markers Nanog and Oct3/4 and an epiblast marker Fgf5 [171]. Conversely, HIF-1α overexpression induced pro-neural Sox1 both directly by binding to *HRE*s in its promoter and indirectly by BMP signaling inhibition [171]. This effect of varying oxygen concentration was specific to differentiating ESCs with no effect on expanding pluripotent cells [171]. Mildly hypoxic culture conditions (4% O_2_) also promoted the survival of NSCs differentiating from ESCs, an effect mediated by apoptosis-inducing factor (AIF) in a HIF-1α-independent manner [172]. However, in contrast to these findings, decreasing HIF-1α during NECs differentiation under hypoxia or in mouse ESC-derived neurospheres decreased self-renewal, promoted neuronal differentiation, and accelerated neurogenesis [173]. In another study normoxia also induced more robust neuroectoderm production [169]. These seemingly contradictory findings may be due to varying effects of oxygen and HIF pathway at various stages of differentiation. When the effects of mild hypoxia (5% O_2_) in vitro were analyzed, opposing outcomes were found during two phases of neural lineage differentiation [174]. First, the differentiation of mouse ESCs into NSCs was inhibited by lower oxygen concentration underscoring the role of hypoxia in maintaining pluripotency. In contrast, secondary differentiation from NSCs to more committed NPCs in neurospheres was enhanced by low oxygen, which also promoted neural over mesodermal or epithelial cell fates [174]. Disparate outcomes of varying oxygen concentration in vitro suggest that the exact timing of hypoxia in relation to the progenitor subtypes during lineage progression, their internal state or subtle changes in medium composition may determine the effect on the proliferation/differentiation balance.

Among NPCs derived from different brain regions mesencephalic progenitors seem to be particularly sensitive to effects of oxygen in vitro [93,175,176,177,178,179]. Culturing rodent or human embryonic mesencephalic NPCs under 2–3% O_2_ instead of normoxia led to more proliferation, promoted pluripotency, reduced cell death and senescence, and enabled higher downstream yield of TH^+^ dopaminergic neurons [93,127,175,176,178,179]. Hypoxic conditions also lead to HIF-1α stabilization [175] and transcriptional activation of its target genes such as *vegf* and *epo* [93]. Increasing HIF-1α levels under normoxia was sufficient to mimic these cellular phenotypes suggesting that it is the main pathway mediating effects of oxygenation on NPCs [127,178]. Consistent with this, the neural progenitor-specific KO of HIF-1α under hypoxic conditions in culture decreased proliferation and increased apoptosis [177]. Moreover, the dopaminergic differentiation and maturation of midbrain NPCs was impaired. This phenotype could be partially rescued by adding VEGF into the medium. The stability of HIF-1α protein in rat mesencephalic NPCs was shown to be dependent on Hsp90 activity. Its inhibition also led to diminished HIF-1α target *vegf* and *epo* mRNA levels, decreased progenitor cell survival and proliferation in vitro [180]. Interestingly, in contrast to mesencephalic progenitors in vitro cortical NPCs did not show significant oxygen level dependent changes in proliferation and pluripotency in these studies [93,177]. However, this lack of oxygen sensitivity may be related to the particular culture conditions or the age and type of isolated progenitors, as other studies reported significant effects of hypoxia on cortical NPCs. For example, hypoxia (2–5% O_2_) in vitro permitted long-term expansion of mouse fetal cortical NPCs [181]. In contrast, normoxia reduced HIF-1α and its anti-apoptotic target nucleophosmin (NPM), while triggering p53 phosphorylation. This led to increased apoptosis, particularly in multipotent progenitors and glial precursors compared to more neuronally committed cells [181]. O4^+^ oligodendrocyte progenitors were especially affected. Rat embryonic cortical NSCs grown under hypoxic conditions (1% O_2_) also increased their proliferation. Under these conditions, HIF-1α expression was increased and mRNA levels of multiple cell adhesion and ECM-related genes were modulated leading to reduced cell adherence and enhanced migration. Among others the MMP-9 levels were increased in a Wnt-dependent way, which was necessary for the observed phenotype [182]. Similarly, both human fetal telencephalon and diencephalon-derived NSCs increased proliferation and reduced apoptosis under mild hypoxia (2.5–5% O_2_) [100,102]. These conditions also enabled efficient neuronal and oligodendrocyte differentiation. However, lower oxygen pressure (1% O_2_ and below) caused deleterious effects on survival with increase in quiescence and altered mitochondrial morphology [100,102]. Mouse embryonic ganglionic eminence derived NSCs in neurosphere culture also proliferated more under mild hypoxia (1–4% O_2_) [183] suggesting that physiological hypoxia generally promotes progenitor expansion and survival in multiple brain regions.

O_2_ tension also affects cell fate decisions during NPC differentiation toward neuronal or glial lineages. Several studies have shown varying effects of hypoxia on neuronal/glial differentiation balance in vitro. Milosevic and colleagues showed that higher oxygen concentrations in culture favored glial over neuronal differentiation in mouse progenitors [127,176]. Similarly, while low oxygen concentration (5% O_2_) promoted proliferation of human NPCs, normoxia increased differentiation into astrocytes [165]. In contrast, another study reported that HIF pathway activation by physiological hypoxia or PHD inhibitors during neuronal differentiation pushed the cells towards GFAP^+^ (glial fibrillary acidic protein-positive) glial fate in a HIF-1/2α and HIF-1β-dependent way, without affecting cell cycle or apoptosis [170]. In addition, increased apoptosis was shown to specifically affect progenitors committed to glial but not neuronal lineages grown under normoxic conditions indicating increased sensitivity of glial lineage cells to high oxygen levels in vitro [181]. In a study by Horie and colleagues, hypoxia (1–4% O_2_) promoted neuronal differentiation into Tuj1^+^ neurons with no apparent effect on the differentiation of GFAP^+^ astrocytes [183]. These seemingly contradictory findings could be explained by subtle variations in the identity of the in vitro propagated progenitor cells or different culture conditions. For example, length of progenitor culture was shown to affect fate choices of mouse ESC-derived NSCs independently of oxygen concentration [184]. Undifferentiated Sox1^+^ NSCs were passaged for short or long period of time and then subjected to normoxic or hypoxic conditions during differentiation. In both cases hypoxia or HIF-1α upregulation reduced stemness and promoted differentiation, however, short passaged cultures preferentially differentiated into neurons, whereas long passaging promoted glial fate [184], indicative of the previously described intrinsic developmental timer [69]. Similarly, cultured NPCs isolated from E15.5 murine embryos could be induced to become astrocytes by LIF, while those from younger E11.5 embryos could not unless pre-cultured for 4 days in vitro [185]. The efficiency of astrocytic differentiation from the younger progenitors was additionally modulated by hypoxia and HIF-1α, which were required for the demethylation of the astrocyte marker genes *gfap* and *S100β*, thereby increasing glia production [185].

The often-discordant results coming from in vitro studies underline the many challenges of investigating the effects of oxygenation and HIF pathway during brain development. While poor specification of progenitor subtypes and lack of control over local variations in neurosphere and organoid cultures can explain some of the discrepancies, their relation to in vivo states are often unclear. In an intact developing fetus, spatiotemporal changes in oxygen concentration brought about by dynamic regional vascularization patterns create a complex local tissue environment. This environment can in turn differently affect various cell types or even their subpopulations i.e., depending on cell cycle phase or at the point of cell fate decisions. Moreover, hypoxia-dependent HIF signaling in these cells can lead to the production of angiogenic and other signaling cues that modify the niche and provide feedback to ingrowing vasculature. Therefore, studies in vivo, which preserve an intact tissue context or its aspects are necessary to better explain the interactions between oxygen tension and HIF signaling in relation to cell type, position in the tissue, internal state and sensing of other external cues.

## 7. The Impact of Hypoxia and HIF Pathway on Neurogenesis In Vivo

The expression of HIF pathway components at the organ level was characterized more than 20 years ago [113,186]. In the brain, the presence of the core transcriptional machinery of the HIF pathway can be detected early-on during CNS development. Mouse embryos express high levels of HIF-1α and HIF-1β mRNA in the E9.5 neuroepithelium, while HIF-2α shows only sparse expression, mainly associated with ingrowing vasculature [113]. This mRNA expression pattern of HIF-1α and HIF-2α continues through later neurogenic stages up until birth, around E19.5 [113,186]. Since the presence of abundant mRNA does not necessarily correspond to high protein levels due to the tight post-translational regulation of HIF-1/2α, it is important to also assess protein expression in the developing brain. HIF-1α protein was shown to be constitutively present in NSCs and NPCs isolated from both the mouse embryonic mid-gestational telencephalon as well as the adult neurogenic zone (P28 SVZ) [187]. In vivo in mouse embryos HIF-1α protein expression could be detected in neuroepithelium as early as E7.5 [173]. It was also detected around mid-gestation in E13.5 embryonic germinal zones [142] and adult SVZ and SGZ (subgranular zone) specifically in nestin^+^ and Sox2^+^ progenitors and GFAP^+^ astrocytes but not in neuroblasts [187]. Human cortical radial glia in the VZ of the dorsal telencephalon also express HIF-1α protein as do young neurons in the CP of a 2nd trimester (14–19 week of gestation) human fetus during upper layer neuron formation [100]. HIF-1α protein in the germinal zone was present in both cytoplasm and nuclei, with the nuclear localization being enhanced by increase in hypoxia [142,187]. Interestingly, in isolated progenitor cells, HIF-1α was protected from hydroxylation and ubiquitination in the cytoplasm even under normoxic conditions despite the presence of PHD and VHL enzymes due to its retention in vesicular structures [187]. Meanwhile, consistent with the mRNA expression pattern, no HIF-2α protein was detected in the embryonic cortex at mid-gestation using Western blotting [142].

The first insights into the in vivo role of the HIF pathway during brain development came from the characterization of the *nestin-Cre*-mediated HIF-1α KO mice [188]. In contrast to global HIF-1α KO animals [139,141], brain-specific HIF-1α null mice are viable. However, they present with severe neurodevelopmental impairments resulting in a reduction in the number of neurons and cortical thickness (without changes in layering), low telencephalic vascular density and hydrocephalus. The observed reduction in neurons was caused at least in part by widespread apoptosis thought to stem from angiogenesis defects and resulting hypoxia [154,188]. As a consequence, HIF-1α mutant mice presented with impaired spatial memory. Cellular and vascular density phenotypes could be rescued by in utero electroporation of the wild-type HIF-1α suggesting its cell autonomous role in progenitor cells and/or their progeny. In this study the first defects were seen perinatally while mid-gestation (E15.5) embryos appeared normal [188]. However, the early effects of HIF-1α deletion on NPCs could have been missed due to low efficiency of *nestin-Cre* mediated recombination before E17.5 [147,189]. To overcome this limitation, Sakai and others [163] deleted HIF-1α in NECs using Sox1-cre mouse line, in which Cre recombinase is expressed in the neuroepithelium as early as E8.5. This resulted in early postnatal lethality, abnormal head shape and brain defects, particularly reduced telencephalon size visible from E14.5. Increased neuronal apoptosis was seen especially in deep-layer neurons concomitant with layer disorganization. *Vegf* mRNA expression was reduced in KO brains and VEGF signaling inhibition correlated with apoptosis in a non-cell autonomous way pointing to disrupted angiogenesis as the cause of this phenotype. In contrast, the positioning of neurons was regulated cell autonomously as HIF-1α KO neurons localized more apically suggesting impaired migration. However, the authors did not observe any major effect on proliferation or NPC numbers, possibly due to the length of time between onset of HIF deletion and analysis [163].

The functional consequences of the local hypoxic environment and HIF pathway activation on NPCs in vivo were demonstrated in an elegant study by Lange and colleagues [142]. The authors genetically manipulated cortical vascular development specifically in the embryonic brain by using Gpr124 KO mice, which exhibit impaired angiogenesis in restricted CSN regions, including dorsal telencephalon. Vascularization defects resulted in prolonged hypoxia, increased HIF-1α stabilization and transcriptional activation of its target genes including Glut1 upregulation. At the cellular level prolonged hypoxia in vivo promoted NCS expansion at the expense of differentiation. Particularly, the proportion of aRG increased, while the number of more differentiated BPs (bRGs and IPs), which are normally generated from aRG by neurogenic divisions [37,38,42,50], diminished in KO embryos [142]. This phenotype was shown to be caused by changes in tissue oxygenation as it could be rescued by transferring pregnant dams to hyperoxic environment (80% O_2_ atmosphere), which relieved hypoxia. The HIF signaling pathway was involved in mediating the effects of hypoxia. HIF-1α deletion specific to NPCs using *Emx1:Cre* increased early neurogenesis at the expense of progenitor expansion resulting in postnatal reduction in cortex size and thickness. No effect on cortical layer organization was observed. However, HIF-1α KO had a pleiotropic effect on the cortical tissue by causing a reduction in cortical angiogenesis and induction of apoptosis, making it difficult to distinguish direct effects of HIF signaling from its effects mediated by vascularization. Nevertheless, in utero electroporation of HIF-1α at E13.5 resulted in an increase in cycling RG and a reduction in cortical neuron number in CP suggesting a direct influence on the proliferation/differentiation balance. This effect was also shown to require the transcriptional activity of HIF-1α [142].

Tissue oxygenation was also shown to regulate the expansion of basal RG progenitors in the embryonic mouse brain [154]. This progenitor type is particularly abundant in the oSVZ of gyrencephalic species, whereas small lissencephalic rodents only have a limited number of bRG in their lateral cortex [190,191]. Systemic exposure of pregnant dams to 10, 21 or 75% oxygen for 48 h during mid-neurogenesis (E14.5 to E16.5) correlated with fetal brain oxygenation and brain growth. Hypoxic embryos had increased apoptosis and decreased tissue volume with thinner CP and fewer upper layer neurons. Conversely, under hyperoxic conditions, brain size and cortex thickness increased while apoptosis was decreased. Histologically hyperoxia led to an expansion of proliferative Sox2^+^ bRG and Tbr2^+^ and Sox2^+^ double positive progenitors, which accumulated basally of normal SVZ range. This in turn led to an increase in the proportion of Ctip2^+^ neurons produced. The effect on bRG cells was also shown to be dependent on intact HIF-1α signaling. Importantly, the effects of tissue oxygenation and HIF pathway activation were not limited to the developing telencephalon but also regulated NPC proliferation and dopaminergic neurogenesis in the embryonic midbrain [99]. Systemic hypoxia or hyperoxia in pregnant females led to changes in embryonic midbrain size. These oxygenation changes correlated with midbrain VZ hypoxia, which was exacerbated in fetuses, in which HIF-1α was conditionally deleted using *nestin-Cre*, as an effect of decreased vessel density. The numbers of dividing cells in the VZ and newly produced neurons in both the SVZ and the mantle zone increased with oxygen concentration. Oxygenation also exerted some HIF-1α-independent effects, as the total neurogenesis of TH^+^ neurons was positively correlated with O_2_ levels regardless of the HIF-1α KO [99].

In addition to the effect on neurogenesis, changes in oxygen tension and HIF pathway activation were also shown to regulate the expansion, survival and maturation of OPCs both in vitro [100,165,181,192] and in vivo [193,194]. The relationship between culture oxygenation and oligodendrocyte generation is complex. Hypoxia has been shown to differentially affect astrocytic and oligodendrocyte progenitors, with OPCs being particularly sensitive to hypoxia-induced cell death [100,181]. Even temporally limited variations in oxygen pressure affect OPC fate. Low in vitro O_2_ concentration during OPC proliferation stage permits robust differentiation after switching to normoxic conditions in vitro. In contrast, exposure of dividing oligodendrocyte precursors to high O_2_ causes p53 phosphorylation and p21cip1-mediated mitotic arrest [165]. Consistent with the effects of oxygenation in culture [192], mild postnatal hypoxia or constitutive HIF-1/2α stabilization by VHL inhibition in vivo caused hypomyelination and delayed OPC maturation in rodent neonates [194,195]. In turn HIF-1/2α KO inhibited angiogenesis, which normally occurs in the corpus callosum between P0–P4, resulting in cell loss and axonal disruption. In vitro, the hypoxic OPC maturation arrest was shown to rely on an autocrine Wnt7a/7b activation via HIF-mediated transcription, with HIF binding directly to *HRE*s in the promoter region [194]. Secondarily, paracrine Wnt activity from OPCs also affected early postnatal white matter angiogenesis in vivo by stimulating EC proliferation [194]. While the stimulatory effect of oligodendroglia on EC proliferation and angiogenesis in various CNS regions in vivo has recently been confirmed, the role of Wnt signaling in this process is disputed [192,193]. Instead, the canonical VHL-HIF-1/2α-VEGFA-pathway has been shown to mediate this effect. Meanwhile, blocking the activation of the Wnt/β-catenin pathway in ECs, which was mediated by HIF-1α activation and downstream Wnt secretion from astrocytes, reduced angiogenesis [193]. Oxygen sensing also interacts with other signaling pathways during glial differentiation. The BMP antagonist noggin has a limited effect on the behavior of glial progenitors in 20% O_2_ culture but it promotes progenitor cell expansion and decreases glial differentiation under hypoxia by inhibiting the BMP/Smad pathway. Conversely, BMP2 increases the number of produced GFAP^+^ glial cells and decreases the number of progenitors only in normoxia [165].

HIF-1α in NPCs as well as in postmitotic neurons can also be induced in a hypoxia-independent manner. Constitutive activation of the metabolic master regulator mTOR kinase in neural progenitors during early neurogenesis triggered both HIF-1α upregulation as well as massive apoptotic cell death in layers basal to VZ [196]. While this activation had no effect on aRG proliferation, the number of Tbr2^+^ cells was significantly reduced suggesting a specific effect on basal progenitors. Whether the loss of bIP cells could be attributed to apoptosis or diminished production from HIF-1α overexpressing aRG [142] was not assessed. Nonetheless, as an effect of this cell loss mTOR hyperactive pups presented with microcephaly and failure to thrive resulting in early death. In contrast, induction of mTOR hyperactivity in postmitotic neurons, either perinatally or in young adults had no effect on apoptosis. Instead, defects in neuronal migration were observed as well as cortical hypertrophy associated with increased neuron body size. In addition, signs of neurodegeneration and massive microglia activation can be observed. One of the possible cues attracting the microglia could be an HIF-1α-dependent increase in extracellular adenosine caused by local tissue hypoxia [197]. These postnatal phenotypes led to severe epilepsy and death in affected mice. It is currently unclear if the postnatal phenotypes were also associated with HIF pathway activation nor is it known to which extent it was responsible for the disease presentation. The possible effects of mTOR-mediated HIF-1α activation of angiogenesis were also not assessed in this study [196].

Most in vivo studies have analyzed the role of HIF signaling during neurogenesis either by targeting HIF-1α specifically or by general upstream modulation of HIF-1/2α via components of the hydroxylation and ubiquitination pathway. Although HIF-2α has received less attention due to the lack of an overt KO phenotype in the brain [198] or expression in neurogenic progenitors [113,142,186], a recent study has shown that it is required for neuronal survival [195]. In contrast to HIF-1α, which is only stabilized under systemic hypoxia, HIF-2α protein is present in the brain of adult wild type mice under normoxia. Mice with a conditional HIF-2α KO using *nestin-Cre* had a normal lifespan and fertility but reduced cortex size and the number of pyramidal neurons specifically in retrosplenial but not PFC or hippocampal areas. This cellular phenotype resulted in functional impairments in learning and memory. Cultured neurospheres from new-born HIF-2α KO NPCs did not reveal differences in proliferation. Instead, the cells showed lower migration under normoxia, suppression of cell death and reduced Neurogranin, Syn1 (Synapsin 1) and Dlgap4 (DLG Associated Protein 4) mRNA expression, suggesting impaired synaptic function. mRNA profiling from KO and WT neurospheres cultured under hypoxia also revealed altered neurogenic pathway gene expression. In addition, HIF-2α KO caused a reduction in MBP (myelin basic protein) mRNA, a marker of mature oligodendrocytes, in vivo [195]. Because the authors only analyzed adult mice it is unclear if the observed phenotypes could be attributed to developmental as opposed to postnatal functions of HIF-2α.

Interestingly, adult neurogenesis in rodents has also been shown to require appropriate physiological oxygen levels and the HIF pathway modulation [199,200]. Resident NCSs/NPCs in both main neurogenic zones of an adult brain, SVZ and DG SGZ, express HIF-1α [201]. Reduced O_2_ pressure in the DG and SVZ due to systemic hypoxia increased HIF-1α and VEGF expression and led to increased NSC proliferation in both regions [94]. HIF-1α KO in NPCs of the adult SGZ, which are responsible for the hippocampal neurogenesis, resulted in a 50% reduction in new-born neurons in DG. This negatively affected hippocampus-dependent cognitive tasks, namely context-dependent fear and operant learning [199]. Similarly, in the adult murine SVZ, which produces neurons for the olfactory bulb [33], HIF-1α deletion using a tamoxifen-inducible *nestin-Cre* model led to reduction in NSC maintenance [200].

In summary, variations in oxygen tension in vivo caused by regional differences in vascularization during embryonic development affect progenitor cell behavior mainly via HIF-1α pathway. Consistent with the bulk of in vitro evidence, the primary role of hypoxia in NPCs seems to be the preservation of stemness and suppression of neurogenesis. In contrast, low oxygen tension has a negative effect on the survival of more differentiated cell types such as OPCs or neurons and the HIF pathway plays a role in the prevention of cell death. The reciprocal regulation of the oxygenation and HIF signaling and their effect on progenitor behavior are summarized graphically in Figure 3A.

## 8. Molecular Mechanisms of HIF Pathway Regulation in Neurogenesis

Studies on the role of hypoxia in NSC and NPC behavior have shown a dominant role of the HIF signaling pathway in mediating its effects on cellular phenotypes. The molecular mechanism of HIF action can be largely explained by its role as a transcriptional activator [105,106,138]. The downstream targets of HIF transcriptional regulation as well as the crosstalk with other signaling pathways are discussed in this chapter.

One of the main functions of the HIF pathway-activated transcription is the promotion of angiogenesis through increased expression of its canonical HIF target *Vegf* [118,152]. Human NSCs in culture treated with HIF-1α stabilizing compounds such as PHD inhibitors increased their VEGF production while decreasing proliferation and increasing dopaminergic differentiation [202]. In vitro embryonic mouse NCSs/NPCs isolated from E14 telencephalon supported morphogenesis of capillaries and had a protective role on ECs preventing cell death after glucose and O_2_ depravation (<0.2%). VEGF signaling was required for this protective function [202]. A trophic effect on co-cultured neurons was also observed [203]. NCSs/NPCs but not ECs in culture constitutively expressed HIF-1α and VEGF under normoxia, which were further upregulated by glucose and O_2_ deprivation [202,203]. A similar pro-survival effect was seen after intracerebral transplantation of these progenitors into the dorsal striatum. Transplanted NCSs/NPCs expressed HIF-1α and secreted VEGF, thereby supporting local capillary growth and offering neuroprotection against O_2_ and glucose deprivation following ischemia [201,203]. Furthermore, NPCs in the adult SVZ were also ischemia resistant, neuroprotective in vitro and expressed HIF and VEGF [203]. In vivo, local hypoxic conditions and HIF-1α stabilization in the pre-neurogenic dorsal telencephalon were shown to be required for correct vascularization largely in a VEGF-dependent manner [142]. Similarly, HIF-induced VEGF expression regulates angiogenesis in the corpus callosum [193] and retina [204]. In the adult SVZ, HIF-1α deletion in NSCs led to reduced VEGF production and impaired vascular stability in the niche [200]. The reduced local vascular density was followed by defects in NSC maintenance and their depletion in a non-cell autonomous manner.

The proangiogenic effect of HIF activation creates a negative feedback loop in which local hypoxia initially induces HIF stabilization, VEGF production and the ingrowth of blood vessels leading to a subsequent increase in oxygenation and HIF degradation (Figure 3B). This makes it difficult to experimentally separate the direct effects of HIF-mediated transcription on cellular phenotypes from secondary effects of vascularization. The combined effect of a vascularization defect and the resulting downstream HIF pathway activation on gene transcription in VZ progenitors was assessed by [142]. Prom1^+^ (CD133) VZ cells from normal and vasculature-defective cortices at E14.5 were sorted and their RNA sequenced. Approximately the same number of genes, about 250, was up- and down-regulated upon vascular defect. Among the upregulated genes angiogenesis and cell proliferation functions were overrepresented, while neurogenic genes were downregulated. The data set included predicted HIF-1/2α targets such as *Vegfa*. In addition, *Tbr2* and *Bmi1* (BMI1 proto-Oncogene, Polycomb Ring Finger) transcripts were decreased in vasculature-defective cortices, indicative of altered neurogenic progenitor differentiation [142]. While a number of these gene are likely direct HIF targets this experimental setup did not allow to distinguish direct from indirect regulation. Timed activation of HIF-1α expression followed by immediate RNA sequencing before angiogenesis is induced as well as ChIP analysis would help to isolate HIF targets from their downstream effects and crosstalk with other pathways. Alternatively, combining HIF-1α deletion with a VEGF rescue could help disentangle vasculature-dependent and independent phenotypes.

HIF-mediated transcriptional regulation also controls cell autonomous metabolic functions. One of the most well-established cellular functions targeted by the HIF pathway is glycolysis [104,138,142,205], which in addition to producing ATP provides intermediates for other metabolic pathways [206]. Rapidly dividing cells such as stem or cancer cells have been described to rely on glycolysis and HIF activation to support their growth [134,206,207,208]. ESCs are known to preferentially use glycolysis [209] and HIF-1α KO in these cells in vitro reduces the transcription of glycolytic enzymes (*Alda*, *Pgk1*, and *Eno1*), lactate dehydrogenase (*Ldha*) as well as glucose transporters (*Glut1* and *Glut3*) [139,140]. Mutant cells are able to survive normally but reduce their proliferation rate [139]. Physiologically HIF-1α appears as a main regulator of glycolysis, but its deletion causes compensatory upregulation of HIF-2α transcription [140].

When the metabolic status of cultured NPCs was studied in terms of oxidative and glycolytic pathway utilization similar effects were observed as in other proliferating cell types [210]. Murine NPCs derived from either E14.5 embryonic or adult neurogenic zones cultured under normoxia had high lactate production, LDH activity and glucose consumption. Consistent with an adaptation to an oxygen-independent metabolic mode NPCs were also able to survive under anoxic conditions (<0.2% O_2_) but in contrast to neurons, were sensitive to selective glycolytic inhibition even when TCA activity was preserved. They were also less able to use galactose, which does not lead to the production of ATP from glycolysis, instead of glucose as an energy source. Meanwhile, neurons were less resistant to mitochondrial electron transport chain inhibition. Moreover, the NPC dependence on glycolysis was not only due to aerobic glycolytic ATP production. Glycolysis has been shown to play a role in feeding the pentose phosphate pathway (PPP) [207,208] and NPCs were also sensitive to PPP inhibition [210]. Surprisingly however, HIF-1α inactivation had only a minor effect on these metabolic characteristics. Although the authors did not observe an increase in HIF-2α protein levels upon HIF-1α deletion a compensatory effect could not be completely ruled out [210]. In contrast, HIF-1α activation was shown to increase the expression of glycolytic enzymes and lactate levels in cultured OPCs [192].

HIF-1α-mediated promotion of glycolytic activity was also described to play a role during embryonic cortex development in vivo [142]. Proliferative aRG (Prom^+^/Tis21^−^) showed upregulation of a number of HIF-1α targets related to glucose metabolism including *Mct4*, *Glut1* and *Pdk1* in comparison to more differentiated neurogenic progenitors (Prom^+^/Tis21^+^). As angiogenesis and oxygenation progressed during development there was a reduction in HIF-1α levels and downregulation of glycolytic genes in both mouse and ferret embryos. In turn, prolonged hypoxia during neurogenesis in the vasculature-defective Gpr124 KO upregulated the transcription of glycolytic enzymes (*Pfkfb3*, *Hk2*, *Gapdh*, *Eno1*, and *Ldha*) and other glucose metabolism-related genes (*Glut1*, *Pdk1*), which correlated with the expansion of proliferative progenitors. Moreover, embryonic NPCs in culture also produced more lactate during the proliferation phase, which decreased upon differentiation. This phenotype was dependent on Pfkfb3 (6-Phosphofructo-2-Kinase/Fructose-2,6-Biphosphatase 3), a rate-limiting enzyme of the glycolytic pathway, which was also highly expressed in VZ cells and regulated by HIF-1α. In vivo silencing of *Pfkfb3* increased neurogenesis at the expense of progenitor maintenance downstream of HIF-1α contributing to partial attenuation of HIF-1α overexpression phenotype. Notably, high glycolytic activity in proliferative progenitors did not affect the oxidative phosphorylation, which was also highly active in these cells, at levels comparable with neurons [142]. This phenotype resembles the behavior of other stem and malignant cells which upregulate glycolysis under aerobic conditions [104,134,206,207,208]. Taken together, high glycolytic activity in NSCs promoted by hypoxia and HIF-1α transcriptional regulation is required to prevent premature differentiation in vivo [142].

Another canonical target of the HIF pathway, erythropoietin (EPO) [211], also plays a role in both neural tube closure [212] and NPC proliferation [213], as well as neuronal migration [214] and survival [215,216]. EPO has also been shown to affect the metabolic activity of human NPCs independently of HIF-1α [215], suggesting that some metabolic effects of the HIF pathway can be mediated indirectly through its targets.

In addition to angiogenesis and metabolic regulation by canonical targets known from multiple other cell types HIF pathway activation could specifically affect pluripotency and cell fate specification of NPCs by controlling the expression of key transcription factors. Hypoxic HIF-1α stabilization in fetal rats and neonatal pups coincides with an increase in Pax6 expression in the germinal zones in both the cortex and hippocampus [217,218]. Conversely, Tbr2 transcription is negatively affected by hypoxia induced by cortical vascular defects [142]. Whether the expression of these transcription factors in NPCs is directly or indirectly affected by HIF activity is so far unclear. In contrast, direct HIF pathway targets were studied in cultured mouse OPCs [192]. HIF-1α accumulation was induced by VHL KO in these cells leading to delayed oligodendrocyte maturation. A genome-wide DNA binding profile of these HIF-1α overexpressing OPCs using ChIP revealed two distinct sets of targets. In addition to canonical HIF target genes such as *Vegfa*, *Pdk1* and *Bnip3*, OPC-specific non-canonical targets such as maturation inhibiting *Ascl2* (achaete-scute family bHLH transcription factor 2) and *Dlx3* (distal-less homeobox 3) genes were activated by HIF signaling. This study also revealed that an OPC marker and transcription factor Olig2 can directly interact with HIF-1α protein on OPC-specific target promoters due to a combinatorial code of HIF and Olig2 binding sites at the DNA level. This study also uncovered a secondary, indirect HIF-1α effect on OPC development. OPC-specific HIF-1α targets Ascl2 and Dlx3 bound to the Sox10 promoter and acted to reduce its expression thereby inhibiting oligodendrocyte maturation. A similar effect was mediated by another HIF-1α target in OPCs—a histone deacetylase Sirt1 [219]. Sirt1 expression was upregulated in white matter OPCs upon neonatal hypoxia and HIF-1α stabilization leading to increased proliferation and impaired maturation of these precursors. This effect was partly mediated by deacetylation of Cdk2 (cyclin-dependent kinase 2) and Rb (retinoblastoma) in these cells, which allowed for Rb/E2F1 complex dissociation and cell cycle progression.

The HIF pathway has also been described to interact with other signaling pathways relevant to neurogenesis, particularly Notch and Wnt (Wingless-type) [138] (Figure 3C). The role of hypoxia in maintaining cells in an undifferentiated state depends on both HIF and Notch signaling. When cultured primary embryonic rat NSCs were differentiated into neurons under hypoxia (1% O_2_) compared to 21% O_2_ their differentiation was reduced [137]. However, inhibition of the Notch pathway blocked this effect. The authors showed that hypoxia, acting through the upregulation of the HIF-1α protein, triggered the stabilization of Notch ICD and the transcription of Notch target genes, including *hes-1*. Notch ICD was also able to physically interact with HIF-1α thereby recruiting it to Notch-responsive promoters during hypoxia to co-activate Notch signaling [137]. Conversely, HIF-1α KO in neurosphere culture downregulated Hes-1 independently of Notch signaling. HIF-1α was shown to directly bind to distal *HRE* in *hes-1* promoter to trigger its expression in mouse neuroepithelium between E7.5–E9.5 [173]. Both Notch and HIF could also be regulated by asparaginyl hydroxylation via FIH-1/HIF1AN in hypoxia [220] and Notch ICD was shown to directly bind to FIH-1 [137]. Notch1-3 ICD could be hydroxylated on conserved asparagine residues, which are necessary for its function in preserving stemness of NPCs [220]. Since Notch ICD was shown to have a higher binding affinity to FIH-1 than HIF-1α, it may sequester it away and thus indirectly increase HIF-mediated recruitment to *HREs* and transcription under normoxia (Figure 3C). The interplay between HIF and Notch signaling also plays a role in epigenetic regulation of astrocytic differentiation by affecting *gfap* promoter methylation [185]. While NPCs isolated from mouse embryonic telencephalon after mid-neurogenesis (E15.5) could efficiently differentiate into astrocytes, younger progenitors from E11.5 cortices showed *gfap* gene methylation and low differentiation potency. Under hypoxic growth conditions a Notch target Nfia (nuclear factor IA) could induce *gfap* demethylation. Similar to the observations in NPCs [137], HIF-1α upregulation by hypoxia promoted Notch activation in order to induce astrocytic fate both in vitro and in vivo [185].

Wnt pathway has also been shown to interact with HIF signaling in the context of neurogenesis. Under hypoxic conditions in vitro Wnt/β-catenin signaling is enhanced in both mouse ESCs and NSCs, but not in differentiated cells [221,222]. This coincides with HIF-1α stabilization and nuclear translocation, HIF-1α/ARNT complex formation and its target transcription (e.g., *Pgk1*) as well as increased cyclin D1 (a canonical Wnt target) level and NSC proliferation. Inhibition of either Wnt or HIF-1α pathway abolished the hypoxia-dependent increase in cell survival and proliferation. HIF-1α was shown to directly bind to promoter regions of Wnt pathway-related transcription factors Lef-1 and Tcf-1 via *HRE*s. This led to increased levels of the LEF/TCF complex, which enhances β-catenin nuclear translocation downstream of Wnt pathway stimulation and induces Wnt/β-catenin target transcription [221] (Figure 3C). Conversely, HIF-1α KD decreased nuclear translocation of β-catenin and cyclin D1 expression, which suppressed NPC proliferation [222]. In vivo Wnt pathway activity was shown to colocalize with hypoxic regions in both embryonic and adult brains, particularly in adult SGZ in the hippocampus. SGZ also had fewer blood vessels compared to surrounding regions and expressed HIF-1α regulated genes such as *Vegf* and *CAIX* (carbonic anhydrase IX) [221]. Postnatal neuron-specific HIF-1α deletion in vivo caused reduction in Wnt pathway activity in the adult brain as well as decreased progenitor proliferation, adult neurogenesis and neuronal projections. These phenotypes could be rescued by Wnt pathway activation. Taken together the effect of HIF-1α pathway on neurogenesis seem to be partially mediated by its ability to enhance Wnt signaling, particularly in adult NPCs [221]. However, the relevance of this mechanism for embryonic cortical neurogenesis is controversial [142].

Other signaling pathways may also interact with hypoxia and the HIF pathway by modulating their downstream effectors. For example, MEK/ERK inhibition increases Sox10 expression, which rescues hypoxic OPC maturation arrest downstream of HIF [192]. Maternal undernutrition, which has been associated with the induction of the HIF-1α pathway via the mTOR (mammalian target of rapamycin) signaling pathway [169,223,224]. mTOR was shown to be able to induce HIF-1α expression independently of hypoxia [223]. This mechanism could provide a coordinated metabolic response to both oxygen and nutrient deprivation which coincide in pathological conditions such as tissue ischemia. Figure 3B illustrates the current understanding of the downstream targets of hypoxia and HIF signaling, which regulate NPC behavior. The crosstalk with other signaling pathways is shown in Figure 3C.

## 9. Open Questions and Directions for Future Research

The HIF signaling pathway, as a master regulator of oxygen responsiveness, is not only an essential regulator of angiogenesis and glucose metabolism, but also a key player in the self-renewal of stem cells in various physiological contexts. Activation of the HIF pathway has also been shown to promote the growth and proliferation of cancer cells, at least in part through the hijacking of stem-cell like metabolic profile [134,207,208]. This includes the reliance on glycolysis as the primary source of energy even under aerobic conditions, first described as the Warburg effect [134,206,207,208]. NSCs appear to be no exception to this general principle. Early embryonic NSCs tend to reside in hypoxic cell niches, which promotes the HIF-mediated metabolic regulation of glycolytic enzymes [96,98,142,148,206]. This metabolic shift is thought to be responsible for their sustained pluripotency and ability to self-renew. Early on in development hypoxic conditions prevent premature differentiation but the effects of HIF activation are self-limiting as the secretion of its target VEGF causes the blood vessel ingrowth, relief of hypoxia, and degradation of HIF-1α [142]. Adult neural stem cells also reside in a predominantly hypoxic environment and their reactivation can be promoted by pathological conditions associated with the depletion of oxygen [187,199,200,221]. Despite a plethora of evidence supporting this general mechanism, many questions remain open. The manner in which glycolysis dependence underlies stemness and high proliferative capabilities is not fully understood. Although this metabolic aspect has been extensively studied in the context of cancer cells [134,207,208], the understanding how metabolism controls stemness in normal cells could elucidate potential differences and lead to more targeted and less toxic therapies. Metabolomic studies of NSCs in vivo in the presence and absence of HIF activation could help to shed light on this issue.

Both embryonic and adult NSCs show basal HIF-1α stabilization [142,187] and this molecular feature is also present in the notoriously difficult to treat high-grade gliomas [225]. Given that adult NSCs constitute a likely source of tumor-initiating cells, understanding the regulation of HIF in this population and the source of its oncogenic potential is crucial for development of effective treatment strategies. Similarly, the function of HIF family members in neighboring cells, such as ECs or immune cells, may influence how brain tumors respond to such therapies. Current treatments targeting the HIF pathway in other tumor types deal with issues of specificity and toxicity [104,105,135,225]. Selecting downstream, neural-specific targets of HIF regulation may provide a more precise approach resulting in fewer systemic side effects. A number of genes have been shown to be regulated by the HIF pathway during neurogenesis [142], but a comprehensive transcriptomic study is still lacking. Furthermore, the coupling of HIF activation and angiogenesis makes it difficult to distinguish between direct regulation and secondary effects. An analysis of genomic binding sites in specific neural stem and progenitor cell populations could help to answer these questions. Furthermore, the specificity of the HIF pathway effectors in different progenitor subtypes as well as their neuronal and glial progeny is not fully understood. While in NSCs the HIF pathway activation mediates the proliferative phenotype, in neurons it seems to be primarily necessary for cell survival. The role of molecular players upstream and downstream of HIF in this differential response as well as the interplay with changing epigenetic landscape need to be understood to provide a full picture of this pathway during neurogenesis. Finally, the HIF pathway contributes to the pathologies of fetal and early postnatal development caused by abnormal oxygen levels or vascular defects. Studying the long-term consequences of early insults should provide an opportunity to develop clinical intervention strategies. Here particularly, the importance of in vivo studies using animal models cannot be overstated.

## Figures and Tables

**Figure 1 cells-13-00621-f001:**
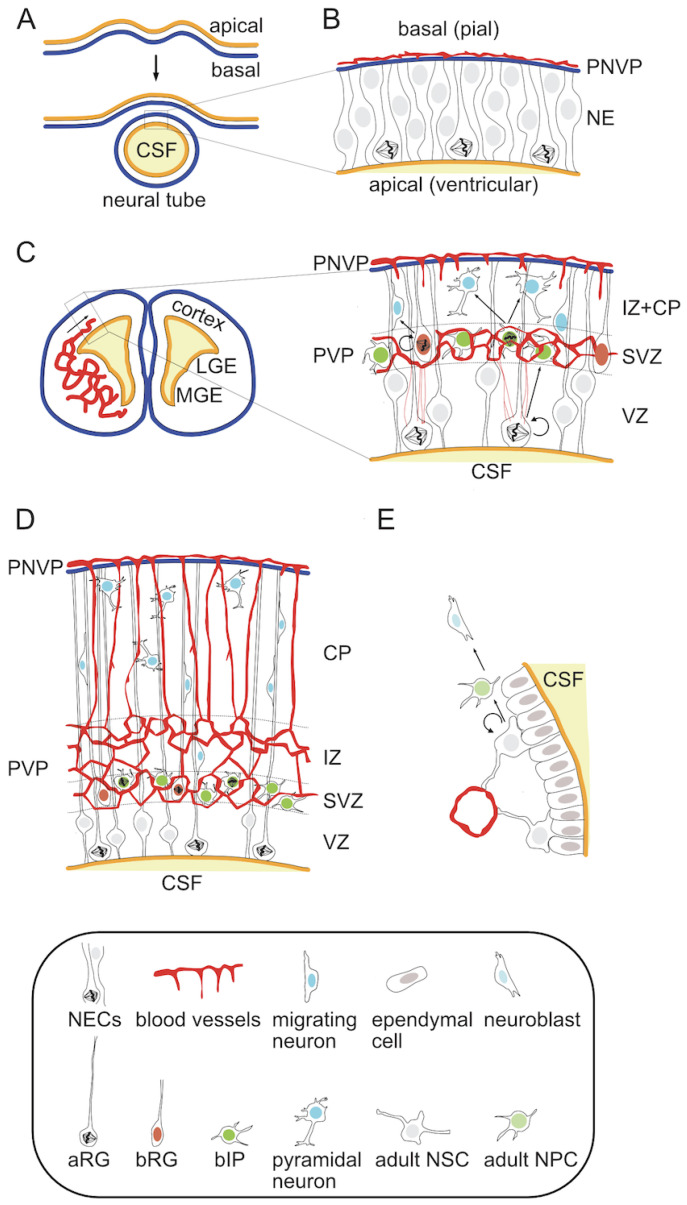
Neurogenesis in the developing neocortex follows a precisely regulated spatiotemporal order of events. During this process progenitor proliferation, neuronal generation and migration coincide with the ingrowth of new blood vessels from two independent sources. (**A**) Neurogenesis is preceded by the formation of a neural tube by the closure of neural folds. This leads to the reversal of the apicobasal polarity with the apical side of the neuroepithelium now facing the ventricular lumen filled with cerebrospinal fluid. The neocortex forms from the dorsal region of the telencephalic vesicle. (**B**) At this stage the neural stem cells are organized in a single layer avascular neuroepithelium. NECs divide symmetrically to expand its surface. Blood vessels are restricted to a pial perineural vascular plexus (PNVP) and do not yet penetrate the brain parenchyma. (**C**) At the onset of cortical neurogenesis NECs transform into aRG, which start dividing asymmetrically to produce more committed basal progenitors and neurons. These migrate basally to form a secondary germinal zone (SVZ) followed by the formation of a cortical plate (CP) by neurons migrating through the intermediate zone (IZ). While the aRG-containing VZ remains largely avascular, angiogenesis occurs in the more basal zones. The PNVP starts sprouting radial vessels through the CP towards the ventricular surface. In the same time a second vascular plexus, the periventricular vascular plexus (PVP) is formed at the border between VZ and SVZ due to the ingrowth of vessels from the ventral part of the brain. This plexus is organized in a honeycomb pattern, with bIPs located preferentially at vessel branch points. (**D**) Both plexuses form connecting branches leading to a singular vascular network as neurogenesis progresses and CP thickens. (**E**) In the germinal zones of an adult neocortex the NSCs reside close to the ventricle and maintain contact with nearby blood vessels. They remain largely quiescent, but can be activated to produce more committed NPCs and neuroblasts. Physiologically, in mice the newly generated neurons in the SVZ migrate along the rostral migratory stream to replenish the neurons of the olfactory bulbs.

**Figure 2 cells-13-00621-f002:**
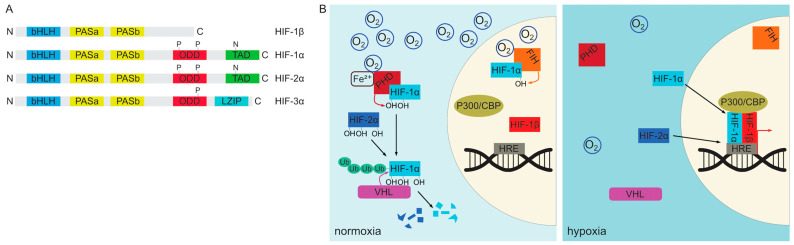
Schematic representation of the HIF protein family members and the molecular mechanism of oxygen sensing via the HIF signaling pathway. (**A**) All HIF family proteins contain an N-terminal DNA-binding basic helix-loop-helix (bHLH) domain and two PAS dimerization regions. The α-subunits additionally contain an oxygen-dependent degradation (ODD) domain. While HIF-1α and HIF-2α have a C-terminal transactivation domain (TAD), HIF-3α has a leucine zipper. Alternative truncated versions of these proteins can also be expressed. (**B**) α subunits of the HIF complex are subject to oxygen-dependent ubiquitination and proteasomal degradation. In the presence of oxygen PHD family enzymes hydroxylate conserved Pro residues in the ODD, which directs HIF proteins to polyubiquitination by VHL. HIF-1α can also be additionally inhibited by asparagine hydroxylation by FIH-1 under normoxia. In contrast, under hypoxic conditions PHD and FIH-1 are inactive, leading to the stabilization and nuclear translocation of HIF-1α and HIF-2α proteins. In the nucleus they can bind the constitutively expressed HIF-1β subunit at the promoters of regulated genes via conserved hypoxia-responsive element (*HRE*) sequences. The complex can then recruit other transcriptional activators to drive the transcription of target genes.

**Figure 3 cells-13-00621-f003:**
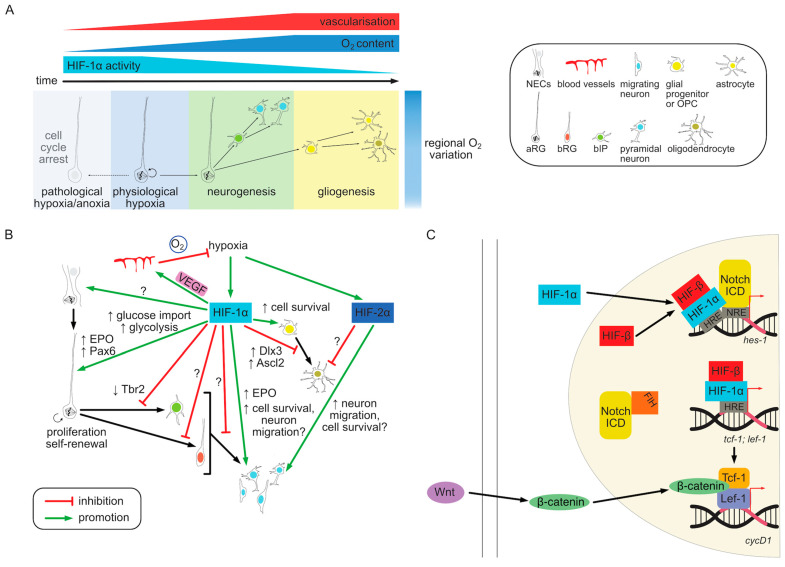
Mechanism of oxygen sensing and HIF pathway activation in embryonic neurogenesis. (**A**) During early development, oxygen delivery to the embryonic neocortex is limited by the lack of vasculature. As a result, HIF-1α signaling is high in aRG cells, which promotes self-renewal. As neurogenesis progresses the ingrowth of blood vessels basally from VZ creates more oxygenated zones where more committed progenitor cells reside. These progenitors divide to produce postmitotic progeny, although its ultimate fate (neuronal or glial) appears to be regulated primarily by the developmental timing. The oxygen levels are also not uniform along the apicobasal axis, which adds another layer to the niche complexity. Although aRG cells reside in physiologically hypoxic regions, very low oxygen levels or anoxia can have adverse effects on progenitor proliferation and survival. (**B**) The effects of HIF-1α activity on different cell types in the developing neocortex. Activation of the HIF pathway generally maintains stemness and promotes survival and proliferation of aRG cells. This occurs at the expense of differentiative divisions and the production of more committed basal progenitors. This phenotype is caused at least in part, by increased glycolysis and HIF-dependent expression of trophic proteins such as EPO. Additionally, HIF promotes the expression of transcription factors such as Pax6 while being associated with the repression of differentiation markers such as Tbr2. However, the degree to which these genes are directly regulated by HIF is largely unknown. Moreover, the role of HIF in regulating basal progenitor subtypes is so far not understood. In newly generated neurons HIF-1α is critical for survival and may affect migration. HIF-1α also induces VEGF production by NPCs which is necessary for cortical vascularization. In glial progenitors HIF activity is particularly important for survival and proliferation of OPCs, while putting the brakes on oligodendrocyte maturation via Dlx3 and Ascl2 regulation. While HIF-2α is unlikely to be required for embryonic NPCs it plays a role in perinatal neuron survival, myelination and likely in ECs. (**C**) HIF signaling cross talks with other signaling pathways during neurogenesis. HIF-1α co-activates the transcription of Notch target genes such as *hes-1* by binding to *HRE* elements directly as well as interacting with the Notch ICD. In addition, Notch can potentiate HIF-1α activity by sequestering FIH-1. HIF-1α also enhances Wnt signaling by inducing the expression of Tcf-1 and Lef-1 transcription factors, which bind β-catenin to activated Wnt-dependent transcription.

## Data Availability

Not applicable.

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
