# Peer review of "From Vessels to Neurons—The Role of Hypoxia Pathway Proteins in Embryonic Neurogenesis"

_cells, 2024, doi:10.3390/cells13070621_

Round 1

Reviewer 1 Report

Comments and Suggestions for Authors

This is a well-written and comprehensive review on the complex impact of hypoxia and hypoxia-inducible factors on the developing brain. The authors concisely present the literature and discuss the challenges of the field.

Author Response

We would like to thank the reviewer for the nice and positive comment on our work.

Reviewer 2 Report

Comments and Suggestions for Authors

This is a carefully-written, well-tailored and informative review covering the role of oxygen and HIFs in embryonic neurogenesis. It is a thoughtful introduction and a significant contribution to the fields of both hypoxia and neural development.

Minor corrections: Lines 1025-1035 must be deleted, they contain reference examples. Disclaimer appears twice, both before and after references.

Author Response

We would like to thank the reviewer for the nice and positive comment on our work.

  • Minor corrections: Lines 1025-1035 must be deleted, they contain reference examples. Disclaimer appears twice, both before and after references.

According to the reviewer’s suggestion we removed the lines 1025-1035 containing the reference examples and the duplicated disclaimer.

Reviewer 3 Report

Comments and Suggestions for Authors

In their manuscript, Stepien and Wielockx review the role of oxygen levels and HIF regulation in cortex development. This is a well-written and comprehensively researched article that provides an insightful review on hypoxia and blood vessel – neuronal cell interactions in the development of the cortex.

The first sections provide a comprehensive review of cortical neurogenesis and vascularisation, hypoxia and hypoxia-sensing, and the interaction between blood vessels and neural progenitors. The section on studies using cultured NPCs highlights the valuable findings about oxygen levels influencing proliferation, apoptosis, survival, differentiation, and cell fate choices, but also discusses the dependence of the results on the cells used and the specific culture conditions. The authors emphasise the importance of complementing the (often contradictory) findings from in vitro experiments with work in vivo, and dedicate the longest section of the manuscript to these experiments. This leads to the section on molecular mechanisms, particularly the transcriptional targets of HIF and the interaction with other pathways. The final section is entitled “Open Questions and Directions for Future Research” – besides open questions around the role of glycolysis and the identification of cell-specific HIF targets, it focuses on the relevance of the hypoxia-neurogenesis link for understanding and treating brain cancers. This could be reflected in the section title, and possibly the abstract.

Specific minor comments and corrections:

Line 31/32: Only a minor point, but it might be worth mentioning that, while the 6-layered neocortex as such is only found in mammals, homologous structures like the dorsal cortex and the Wulst exist in reptiles and birds (see e.g. https://doi.org/10.1098/rstb.2015.0060 for a review on the evolutionary origin of the neocortex). This is relevant, because it widens the range of possible experimental models to address specific questions.

Line 59/60: ‘reversal of polarity’ is somewhat misleading, since the neuroepithelial cells maintain their polarity – just that ‘outside’ is now the neural canal.

Line 78/78: it might be worthwhile stating that this mode of development is specific for the cortex.

Line 202: Could in addition cite e.g. https://doi.org/10.1016/j.molcel.2008.04.009 to avoid solely self-citations.

Line 209/210: References 106, 109 and 110 are not really the best references for this sentence (the focus of 106 is on the hydroxylases, and 109/110 are about HIF-3α). References like 112 should be cited instead.

Line 394-396: Interesting difference between rodents and humans. Could this be related to the potential of dorsal progenitors to generate inhibitory interneurons in humans (https://doi.org/10.1038/s41586-021-04230-7)?

Line 429/430: Other references (e.g. https://doi.org/10.1093/cercor/bhy082, https://doi.org/10.1016/j.neuron.2004.09.028, https://doi.org/10.1523/jneurosci.4162-06.2006) might be informative, and also indicate key molecules involved in the interaction (VEGF-A, Neuregulin-1 and CXCL12).

Line 699/700: Could this be linked to increased extracellular ATP/ADP under hypoxia (review https://doi.org/10.1007%2Fs00109-012-0988-7), and consequently the attraction of microglia?

Line 741: low oxygen tension in has a negative effect

Line 892: revealed that an the OPC marker

Line 895: Which specific HIF-1α targets regulate Sox10?

Line 901: partly mediated by deacetylation of the Cdk2

Line 914: neurosphere culture caused downregulated downregulation of Hes-1 independently

Line 954: For example, the MEK/ERK inhibition increases

Section 8: Should point to Figure 3C in the relevant paragraphs on Notch and Wnt signalling (in addition to the sentence in line 962/963).

Line 989/991: Given that adult NSCs constitute a likely source of tumor-initiating cells, understanding the regulation of HIF

Author Response

We would like to thank the reviewer for the nice and insightful comment on our work and the useful suggestions.

  • Specific minor comments and corrections:
  • Line 31/32: Only a minor point, but it might be worth mentioning that, while the 6-layered neocortex as such is only found in mammals, homologous structures like the dorsal cortex and the Wulst exist in reptiles and birds (see e.g. https://doi.org/10.1098/rstb.2015.0060 for a review on the evolutionary origin of the neocortex). This is relevant, because it widens the range of possible experimental models to address specific questions.

We thank the reviewer for this comment. We modified the sentence to add additional information as follows:

‘It has a recent evolutionary origin, having first appeared in the evolution of mammals [2, 4-7] although homologous structures such as dorsal cortex or Wulst also exist in reptiles and birds respectively [8].’

(now lines 32-33)

  • Line 59/60: ‘reversal of polarity’ is somewhat misleading, since the neuroepithelial cells maintain their polarity – just that ‘outside’ is now the neural canal.

We have modified the sentence to clarify the point raised by the reviewer:

‘This morphogenetic process leads to a change in tissue arrangement so that the apical surface of the neuroepithelium lines a fluid-filled inside of the NT and its basal side faces the outside.’

(now lines 61-62)

  • Line 78/78: it might be worthwhile stating that this mode of development is specific for the cortex.

We now added the word ‘cortical’ to clarify:

‘At the onset of cortical neurogenesis NECs transform into aRG, which start dividing asymmetrically to produce more committed basal progenitors and neurons.’

(line 81)

  • Line 202: Could in addition cite e.g. https://doi.org/10.1016/j.molcel.2008.04.009 to avoid solely self-citations.

According to the reviewer’s suggestion we added the reference.

  • Line 209/210: References 106, 109 and 110 are not really the best references for this sentence (the focus of 106 is on the hydroxylases, and 109/110 are about HIF-3α). References like 112 should be cited instead.

According to the reviewer’s suggestion we have added new references.

  • Line 394-396: Interesting difference between rodents and humans. Could this be related to the potential of dorsal progenitors to generate inhibitory interneurons in humans (https://doi.org/10.1038/s41586-021-04230-7)?

We thank the reviewer for this interesting suggestion. The difference in the ability to generate interneurons in the ventral vs dorsal forebrain could indeed be one of the possible consequences of this evolutionary difference in development. We have now added a speculative sentence to this part:

‘Another possibility is that this morphological difference is linked with the ability of human dorsal progenitors to generate inhibitory interneurons, a function normally restricted to ventral progenitors in rodents [158].’

(now lines 407-410)

  • Line 429/430: Other references (e.g. https://doi.org/10.1093/cercor/bhy082, https://doi.org/10.1016/j.neuron.2004.09.028, https://doi.org/10.1523/jneurosci.4162-06.2006) might be informative, and also indicate key molecules involved in the interaction (VEGF-A, Neuregulin-1 and CXCL12).

According to the reviewer’s suggestion we have added the new references.

  • Line 699/700: Could this be linked to increased extracellular ATP/ADP under hypoxia (review https://doi.org/10.1007%2Fs00109-012-0988-7), and consequently the attraction of microglia?

We thank the reviewer for this interesting suggestion. Indeed, the change in ATP/ADP ratio could serve as an attractive cue for migrating microglia. We have added a new sentence to express that hypothesis:

‘One of the possible cues attracting the microglia could be a HIF-1α-dependent increase in extracellular adenosine caused by local tissue hypoxia [197].’

(now lines 709-711)

  • Line 741: low oxygen tension in has a negative effect
  • Line 892: revealed that an the OPC marker

We have corrected the highlighted mistakes (now lines 751 and 903).

  • Line 895: Which specific HIF-1α targets regulate Sox10?

According to the reviewer’s suggestion we have named the specific targets. Now the sentence reads as follows:

‘OPC-specific HIF-1α targets Ascl2 and Dlx3 bound to the Sox10 promoter and acted to reduce its expression thereby inhibiting oligodendrocyte maturation.’

(lines 907-908)

  • Line 901: partly mediated by deacetylation of the Cdk2
  • Line 914: neurosphere culture caused downregulated downregulation of Hes-1 independently
  • Line 954: For example, the MEK/ERK inhibition increases

We have corrected the highlighted mistakes (new lines 912, 925 and 966).

  • Section 8: Should point to Figure 3C in the relevant paragraphs on Notch and Wnt signalling (in addition to the sentence in line 962/963).

We have added the references to Figure 3C in Section 8 (new lines 916, 933 and 954).

  • Line 989/991: Given that adult NSCs constitute a likely source of tumor-initiating cells, understanding the regulation of HIF

We have added the comma after ‘tumor-initiating cells’

Reviewer 4 Report

Comments and Suggestions for Authors

The present review addresses the importance of the different levels of oxygen found in diverse areas of the developing brain, and how this contributes to the correct formation of different cell types forming it. Additionally, it thoroughly reviews the various signaling pathways that act as oxygen sensors and how this translates into growth and differentiation actions, with emphasis on the hypoxia-inducible factor (HIF) pathway. The review lists studies conducted  in cell cultures and in vivo models, concluding that in vitro models can yield disparate results depending on difficult-to-control factors such as the differentiation stage of the cultured cells. Citing primarily in vivo models, the review explains that in addition to regulating neuron differentiation, pathways initiated by hypoxia also control the differentiation of glial cells in the central nervous system, namely oligodendrocytes and astrocytes. The review is lengthy due to its wealth of information, and although it may become dense in some sections, in my opinion, the provided information compensates for that density. Despite the density of content, the review is very well-written. In my opinion, it is ready for publication once a few grammatical errors are corrected, which are listed below. I would like to congratulate the authors for their effort in compiling this information, which will undoubtedly be of great help to researchers in the field of hypoxia and brain development in general.

Below, I enumerate a few conceptual or grammatical errors I have detected while reading. However, I recommend carefully reviewing the text as there may be more that I have not identified.

- Line 143: Mental retardation or seizure disorders are not clinical symptoms. 

- Lines 247-248: It says "reside," it should be "residue." 

- Line 324: It says "proceeds," it should be "precedes." 

- Line 400: Put a comma after "human evolution?" 

- Line 409: Put a comma after "delaminating BPs?" 

- Line 741: It says "tension in has," it should be "tension has."

Comments on the Quality of English Language

Few minor corrections

Author Response

We would like to thank the reviewer for the nice and insightful comment on our work.

  • Below, I enumerate a few conceptual or grammatical errors I have detected while reading. However, I recommend carefully reviewing the text as there may be more that I have not identified.

- Line 143: Mental retardation or seizure disorders are not clinical symptoms. 

- Lines 247-248: It says "reside," it should be "residue." 

- Line 324: It says "proceeds," it should be "precedes." 

- Line 400: Put a comma after "human evolution?" 

- Line 409: Put a comma after "delaminating BPs?"

- Line 741: It says "tension in has," it should be "tension has."

We have corrected the errors as suggested.